# Keto-on-the-Clock: A Survey of Dietetic Care Contact Time Taken to Provide Ketogenic Diets for Drug-Resistant Epilepsy in the UK

**DOI:** 10.3390/nu13082484

**Published:** 2021-07-21

**Authors:** Bridget Lambert, Kathryn Lightfoot, Rachel Meskell, Victoria J. Whiteley, Kirsty J. Martin-McGill, Natasha E. Schoeler

**Affiliations:** 1Vitaflo (International) Ltd., 182 Sefton Street, Liverpool L3 4BQ, UK; bridgetlambert@vitaflo.co.uk; 2Leeds Children’s Hospital, 2nd Floor, 2 Park Lane, Leeds LS3 1ES, UK; kathryn.lightfoot@nhs.net (K.L.); r.meskell@nhs.net (R.M.); 3Royal Manchester Children’s Hospital, Oxford Road, Manchester M13 9WL, UK; victoria.whiteley@mft.nhs.uk; 4School of Health and Society, University of Salford, 43 The Crescent, Manchester M5 4WT, UK; 5Department of Clinical Sciences and Nutrition, University of Chester, Parkgate Road, Chester CH1 4BJ, UK; k.martinmcgill@chester.ac.uk; 6UCL Great Ormond Street Institute of Child Health, 30 Guildford Street, London WC1N 1EH, UK

**Keywords:** ketogenic diets, dietitian, drug-resistant epilepsy, patient contact, time, dietetic management

## Abstract

Medical ketogenic diets (KDs) are effective yet resource-intensive treatment options for drug-resistant epilepsy (DRE). We investigated dietetic care contact time, as no recent data exist. An online survey was circulated to ketogenic dietitians in the UK and Ireland. Data were collected considering feeding route, KD variant and type of ketogenic enteral feed (KEF), and the estimated number of hours spent on patient-related activities during the patient journey. Fifteen dietitians representing nine KD centres responded. Of 335 patients, 267 (80%) were 18 years old or under. Dietitians spent a median of 162 h (IQR 54) of care contact time per patient of which a median of 48% (IQR 6) was direct contact. Most time was required for the classical KD taken orally (median 193 h; IQR 213) as a combined tube and oral intake (median 211 h; IQR 172) or a blended food KEF (median 189 h; IQR 148). Care contact time per month was higher for all KDs during the three-month initial trial compared to the two-year follow-up stage. Patients and caregivers with characteristics such as learning or language difficulties were identified as taking longer. Twelve out of fifteen (80%) respondents managed patients following the KD for more than two years, requiring an estimated median contact care time of 2 h (IQR 2) per patient per month. Ten out of fifteen (67%) reported insufficient official hours for dietetic activities. Our small survey gives insight into estimated dietetic care contact time, with potential application for KD provision and service delivery

## 1. Introduction

Medical ketogenic diets (KDs) are a group of high fat, adequate protein, and very low carbohydrate regimens comprising five variants designed to induce ketosis: the classical KD [1], medium chain triglyceride KD (MCT KD) [2], modified Atkins diet (MAD) [3], low glycaemic index treatment (LGIT) [4], and the modified ketogenic diet (MKD) [5]. KDs are the treatment of choice for certain neurometabolic disorders, such as glucose transporter type 1 deficiency syndrome and pyruvate dehydrogenase complex deficiency, and are an effective treatment option for drug-resistant epilepsy (DRE) [6,7].

KDs are a resource-intensive treatment, with a dietitian as an essential member of the multidisciplinary team (MDT) [6,8]. Regimens are demanding of the skill and expertise of the dietitian due to their inherent complexity, the need for individualised dietary calculation (to take into consideration clinical condition, age, feeding ability, nutritional requirements, educational status, and socio-economic circumstances), meal planning and recipe development, regular patient/caregiver contact, and liaison with other healthcare professionals. On-going follow-up is crucial to fine-tune the diet, manage any adverse side effects, and for its safe and successful discontinuation. Although attrition from KDs occurs (mainly due to poor diet tolerance or lack of efficacy) [7], and this will impact diet duration, recommendations are that KDs for DRE are discontinued if unsuccessful after a minimum of three months and after two years in patients where they have been efficacious [6]. However, there is no maximum duration for the use of KDs in DRE, and 30% of cases continue for longer or return to diet if seizures reoccur after weaning off [6,9].

Despite recognition of the importance of the dietitian in the application of KDs for DRE, few data exist on the time spent fulfilling this role. Implementation and monitoring has been estimated to take between 8.5 to 50 h of dietetic time per patient, depending on KD type, feeding route, and patient age [10,11,12,13,14]. The last decade has seen a 77% increase in the number of KD centres in the UK and Ireland and an over seven-fold increase in the number of individuals with epilepsy treated with a KD [9]. Growing acceptance of the efficacy of the KD for DRE, the potential for increase in patients regarded as eligible for the KD, and reported medical cost-benefit [9,15] warrants investigation of dietetic time and patient numbers involved to inform providers making epilepsy healthcare choices in the future.

The aim of this study was to investigate estimates of time taken by dietitians in the UK and Ireland to implement and manage KDs in patients with DRE. The results may be relevant for application in the delivery of dietetic services for DRE.

## 2. Materials and Methods

An online survey was developed using Microsoft Forms by a consensus group of ketogenic dietitians, as part of the Ketogenic Dietitians Research Network [KDRN]. The survey link was circulated via email to KDRN members working in 39 KD centres across the UK and Ireland. Data were requested only from members who were dietitians concerning patients currently following the KD for management of DRE only (excluding neurometabolic disorders). Data were collected between 3 June and 5 July 2019.

Dietitians were asked to categorise patients depending on feeding route, KD variant, and, if fed via a feeding tube (nasogastric, naso-jejunal, gastrostomy or jejunostomy), by type of ketogenic enteral feed (KEF) received and to give their best estimate (if the exact time was unknown or difficult to quantify) of the average number of hours (to the nearest whole hour) spent as an individual dietitian on a per-patient basis during the KD patient journey of pre-diet preparation, initial 3-month trial, 2-year follow-up period, follow-up beyond 2 years (if provided), and diet discontinuation. All patient-focused activity comprising direct patient contact and indirect patient activity, i.e., care contact time, as defined by the British Dietetic Association (BDA) [16], was to be included.

Dietitians were invited to use free text to identify and describe specific factors that they considered impacted their time spent managing patients with DRE on a KD, for example, certain patient and caregiver characteristics or the availability of support with patient-focused activity (direct and indirect) from dietetic colleagues and members of the KD MDT. No patient data was requested.

Dietitians wishing to participate in the survey were advised to seek approval from the local research and audit team at their KD centre via email and retain for their records.

Data analysis was undertaken using Microsoft 365 Apps for enterprise, Version 2106 (Build 14131.20320 Click to run).

## 3. Results

### 3.1. Demographics

Responses were received from 15 dietitians representing nine UK-only National Health Service (NHS) KD centres. Six (67%) provided a service to children only (aged 0–18 years), one (11%) solely to adults (aged 19 years plus), one (11%) to adolescents and adults (aged 13 years plus), and one (11%) to all age groups.

Data were obtained for 335 patients of which 267 (80%) were 18 years old or under. A total of 59% (198/335) took the KD orally, 36% (120/335) received a KEF via a feeding tube only, and 5% (17/335) used a combination of tube and oral feeding. A total of 56% (189/335) were on a classical KD (all children). Another 39% (130/335) followed the MKD (67 children, 63 adults), and 5% (16/335) followed the MCT KD (8 children, 8 adults).

By stage of KD patient journey, out of 335 patients on the KD, 34 (10%) were in pre-diet preparation and 38 (11%) in an initial three-month trial. A total of 155 (46%) were within the two-year follow-up stage, and 89 (27%) had been followed up for more than two years. Nineteen (6%) were being discontinued from the diet.

### 3.2. Estimated Dietetic Care Contact Time per Feeding Route and KD Variant during the KD Patient Journey

For all feeding routes, KD variants, and KEFs (where applicable), a median of 162 h (IQR 54) of dietetic care contact time was spent per dietitian, per patient, during the KD patient journey (for a maximum of two years follow-up) (Table 1). Of this, a median of 48% (IQR 6) comprised direct patient contact (Table 2).

Patients following the classical KD via a combination of tube and oral feeding required the most dietetic care contact time during the KD patient journey (median 211 h, IQR 172), and those solely tube-fed a proprietary KEF required the least (median 85 h, IQR 61). Median percentage time for direct patient contact was highest for those following the classical KD, MKD, and MCT KDs orally (50%, IQR 25) and lowest for patients on a classical KD given as a combination of tube and oral feeding (38%, IQR 25).

For orally fed patients, the classical KD was estimated to require the most dietetic care contact time per patient (median 193 h, IQR 213), followed by the MKD (median 104 h, IQR 94) and MCT KD (median 95 h, IQR 82).

For KDs given as a KEF solely via a feeding tube, care contact time per patient per dietitian was highest for a blended food KEF (median 189 h, IQR 148) compared to those using a bespoke KEF (median 162 h, IQR 128) or proprietary KEF (median 85 h, IQR 61) during the KD patient journey.

### 3.3. Estimated Dietetic Care Time per Stage of KD Patient Journey for DRE

Median total estimated dietetic care contact time during each of the four stages of the KD patient journey (for a maximum of two years follow-up) for DRE categorised by feeding route, KD variant, and KEF type (if applicable) is shown in Figure 1.

During the pre-diet preparation stage, patients estimated as requiring the greatest amount of dietetic care contact time were those following either the classical KD blended food KEF (median 13 h, IQR 4), classical KD via combined tube and oral feeding (median 13 h, IQR 10), or the classical KD taken orally (median 13 h, IQR 8). Patients following an oral MKD were estimated to need the least dietetic care contact time (median 4 h, IQR 9).

Throughout the three-month initial trial, the blended food KEF took the most care contact time (median 72 h, IQR 20), whilst the least was for those using a proprietary KEF (median 24 h, IQR 30) or a bespoke KEF (median 24 h, IQR 39).

For a maximum follow-up of two years, patients following the classical KD via combined tube and oral feeding were estimated to take the greatest amount of dietetic care contact time (median 144 h, IQR 120), whilst those on a classical KD proprietary KEF (median 48 h, IQR 36) and MCT KD (median 48 h, IQR 36) required the least.

Estimated dietetic contact care time for diet discontinuation and return to usual diet was greatest for a bespoke KEF (median 8 h, IQR 4) and blended food KEF (median 8 h, IQR 4). It was lowest for the MKD (median 4 h, IQR 3).

Of the 15 respondents, 12 (80%) had patients who had been following the KD for longer than two years. The estimated median care contact time for these patients was 2 h (IQR 2) per patient, per dietitian, per month.

Dietetic care contact time was estimated to be higher for all KDS per patient, per dietitian, per month during the three-month initial trial than during the two-year follow-up stage (Figure 2).

### 3.4. Impact of Specific Patient and Caregiver Characteristics on Dietetic Care Contact Time

Patients identified as needing additional care contact time (time not quantified in free text) were infants, toddlers, and young children, particularly during weaning and periods of rapid growth; adolescents and adults with learning difficulties, poor numeracy or literacy skills, or problems using email or the telephone; those with food allergies or intolerances, following self-restricted diets such as veganism, or with poor tolerance of the KD on initiation; patients with complex medical conditions and co-morbidities; and those with multiple-agency involvement (including school, college, respite care facilities) or living in specialist accommodation.

Caregivers supporting patients with KD provision (for example, parents, relatives, or care staff) identified as increasing the dietetic care contact time needed per patient included individuals with learning and communication difficulties; non-English speakers and those with English as a second language, particularly if an interpreter was needed; those with health and mobility issues impacting their ability to cope with the practicalities of the KD, such as weighing and measuring of foods, and meal preparation; and individuals exhibiting challenging behavior or who were anxious and frequently seeking reassurance and those with health beliefs or dietary preferences not aligning with KD evidence-based practice.

### 3.5. Allocated vs. Actual Dietetic Care Contact Time

In the UK NHS, one whole time equivalent (WTE) dietitian is employed for 37.5 h per week [17]. A total of 445 h per week was officially allocated to the 15 respondents for the provision of KD services for 335 patients, equating to 11.85 WTE dietitians with a mean of 28.3 (range 13.5–57.5) patients per WTE dietitian.

Ten of fifteen (67%) respondents considered they had insufficient official working hours to fully undertake all patient-focused activity in conjunction with their requisite staff, service, and self-focused activities as defined by the BDA [16], which includes duties such as line management, student training, continuous professional and service development, and clinical governance. These dietitians reported spending an extra 4 to 13.75 h per week fulfilling all their job-related commitments. However, the time involved fluctuated depending on patient and workplace demands, the latter often influenced by the need to provide holiday, sick, maternity, and study leave cover for dietetic colleagues as required.

Of the 15 respondents, 12 (80%) received help with patient-related tasks (including administration, training, and follow-up) from dietetic colleagues and members of the KD MDT, such as a dietetic assistant or epilepsy-specialist nurse. Eight (53%) dietitians identified use of a ketogenic meal planner, such as the Electronic Ketogenic Manager (EKM) [18], as a time-saver.

## 4. Discussion

Our survey showed that, in the UK, a median of 162 h of care contact time is estimated to be spent per dietitian, per patient with DRE during the KD patient journey. Those following a classical KD taken orally as a combined tube and oral intake or as a blended food KEF were estimated as taking the greatest amount of dietetic care contact time. Two-thirds of survey respondents considered they had insufficient official working time to fully complete all their patient, staff, service, and self-focused dietetic activities. To our knowledge, no previous study has investigated dietetic care contact time in conjunction with consideration of the varied ways in which the KD is administered, albeit our results were estimated and obtained from a small number of survey participants.

Our estimated results are higher than stated in previous reports. MacCraken and Scalisi calculated that for the classical KD, dietetic time was approximately 16 h in total per patient, with diet duration ranging from 5 weeks to 2.75 years [10]. Pre-admission appointments took an average of 55 min for tube-fed patients and 127 min for oral feeders, with inpatient admissions for diet initiation averaging 9 h (range 5.5 to 20). Outpatient follow-up, mostly via telephone, averaged 7 h. In comparison, Lord and Magrath’s UK survey estimated that implementation and monitoring of the classical KD took more than 50 h of dietetic time per patient [11]. Our data suggested that modified KDs require less dietetic time compared to the classical KD, in accordance with previous reports, although our estimations of required time are somewhat higher. Dietetic management of adult patients with glioblastoma receiving the MKD in the UK has been reported to take 8.8 h (4 h non-clinical, 4.8 h clinical) per patient over a 12-week period [12]. In the USA, a minimum of 2.5 h of face-to-face dietetic contact time was needed for adults on a MAD. This included an individual enrollment appointment, group education session (1–1.5 h), and 0.5-h follow-up visits after three and six months and then annually, depending on diet duration [13]. Dietetic time needed for other patient-associated duties were not outlined. Kossoff and Dorwood [14] described initiation of the MAD in an outpatient setting with limited dietetic involvement and dietary education taking approximately 0.5–1 h.

Differences in estimations of dietetic care contact time spent, for any KD variant, will be influenced by the range of patients referred and the level of support they require. Country-specific dietetic practice and differences in medical service funding and resource availability may also impact estimations of time taken, as previous UK data are closer to our findings compared to those from the USA. In addition, differences may exist between the type of patient care contact that was examined, i.e., direct, indirect, or combined, as this will have influenced the total amount of time reported. In addition, several studies have measured the time taken in hours prospectively [10,12,13,14], whereas our results are based on retrospective estimation. Therefore, direct comparisons between our findings and those of other authors may be problematic.

Our results suggest that certain KD variants and feeding routes are potentially more time and resource efficient than others. However, it is important to emphasize that KDs are highly individualised, which prohibits automatic allocation of specific diet prescriptions, standardised initiation protocols, and meal plans. In practice, choice of KD variant is guided not by time taken but by medical and nutritional need alongside the social circumstances, abilities, and preferences of patients and caregivers. For example, a more time-consuming bespoke or blended food KEF may be indicated rather than a proprietary KEF because of food allergies, intolerances, or to specifically meet individual nutritional requirements; a modified KD may be favoured for older children and adults for reasons of palatability and social acceptance rather than the requirement for less dietetic care contact time in comparison with the classical or MCT KD. KD variant selection may also be influenced by clinical evidence, particularly for younger children, where the classical KD has been found more effective than the MAD for patients aged 1–2 years [19].

Individuals with DRE often have significant comorbidities, poor health-related quality of life, and specific socio-economic circumstances that conspire to make comprehending and undertaking KDs challenging both for themselves and their caregivers [20,21]. These factors may directly influence the amount of dietetic care contact time required and will account for high variability between patients, even for the same KD variant. Respondents to our survey identified certain patient and caregiver characteristics they considered necessitated more prolonged and intensive dietetic input leading to an increase in the duration and frequency of care contact time provided or demanded, although the estimated time involved was unquantified. Patient-focused activities described as taking longer were both direct (for example, training and education about the KD, responding to more frequent queries, giving extra reassurance, repetition of verbal instructions, and seeking clarification that information was understood and being followed correctly, especially if an interpreter was involved) and indirect (such as the preparation or adaptation of resources and support materials on an individual patient basis and the sourcing of new nutritional information for specialised food items). Adults and children with multiple-agency involvement required greater effort as, in addition to the immediate caregivers, training, support, and communication was essential to ensure wider health and social care networks were informed and involved. Conversely, once the KD was fully established, respondents indicated that dietetic care contact time was saved by training able patients or caregivers to use an online ketogenic meal planner, such as the EKM [18].

Estimated dietetic time taken per patient will be impacted by the individual patient circumstances or underlying health or social needs and the experience level or individual practice style of the dietitian. It could be assumed that more complex patients or less-experienced dietitians would have reported increased dietetic time, but as we did not collect this data, we cannot confirm this. In addition, external confounding factors are recognised as impacting healthcare service provision, such as facility location, setting, and population served and the skill mix and degree of support available from other members of the MDT [22]. Any one or more of these may have influenced our estimates of care contact time as well as how long the KD service had been established and operational for.

It is of concern that two-thirds of respondents considered they had insufficient official working time to fully undertake all patient-focused activity for their patients with DRE on a KD in conjunction with requisite staff, service, and self-focused activities. This stretched working time may be exacerbated by some patients or caregivers deciding not to start the KD despite pre-diet preparation being completed by the dietitian or by individuals stopping the diet during or at the end of the three-month trial period due to no clinical improvement or the challenges involved in KD implementation; in these cases, the most time-intensive periods will have been undertaken by the dietitian but (usually) without benefit to the patient. It should also be noted that many KD services for DRE are built on the provision of a two-year KD follow-up period, and as such, patients remaining on diet for longer periods increase pressure on resources even if dietetic input needed for these long-term patients is less per month [9].

We fully acknowledge that this study has many limitations. Firstly, despite inviting all 39 KD centres in the UK and Ireland to participate, only 15 dietitians representing 9 UK NHS KD centres completed the online survey. A previous questionnaire circulated amongst the KDRN membership by Whiteley et al., 2020 [9], reported results from 39 KD centres across the UK and Ireland, although the total number of individuals who took part in their survey was not detailed by the authors. Self-selection bias may have influenced the decision to participate in the survey. We have no knowledge about certain characteristics of the fifteen dietitians that took part, as we did not ask for information concerning, for example, level of KD experience, workload, or how many of their patients or associated caregivers required a higher intensity of dietetic input and support. These factors may have also affected reported estimates of time taken for dietetic activities, but we are unable to account for this. Time was estimated retrospectively rather than by a formal prospective workload time analysis. This was a pragmatic approach chosen to place least demand on respondents. The survey was not validated, and therefore the estimates of time we obtained, although they add to the information available on this subject, cannot be relied upon for accuracy nor can they be generalised. Researcher imposition may have occurred given that the survey was designed by a consensus group of ketogenic dietitians. Our survey results are mostly related to paediatric patients, and any difference in time needed for management of children versus adults was not assessed. We did not break down the dietetic care contact time needed by individuals following a KD for longer than two years by KD type, feeding mode, or KEF due to the varying lengths of time that individuals remain on diet. Time required for other healthcare professionals involved in the KD service, such as physicians and clinical nurse specialists, was also not evaluated. Finally, it may be difficult to extrapolate our UK NHS-specific results to other countries.

## 5. Conclusions

Despite its many acknowledged limitations, our survey offers novel insight into the care contact time estimated to be required by dietitians in the UK to manage patients with DRE following KDs considering the variety of modes in which they can be delivered. Subject to careful interpretation and application due to the non-validity of the instrument used for data collection, potential bias from respondents, and data inaccuracy, our results may have application in the identification of inefficiencies in the delivery of patient-focused activities by dietitians during the KD patient journey. Further work could be undertaken to find ways in which the time involved could be reduced or capitalised upon without compromising patient care, for example, during the intensive initiation and three-month trial stages, or to inform the re-design of traditional care pathways for dietary management of DRE and to investigate the relationship between outcomes of patients on the ketogenic diet and dietetic care contact time.

## Figures and Tables

**Figure 1 nutrients-13-02484-f001:**
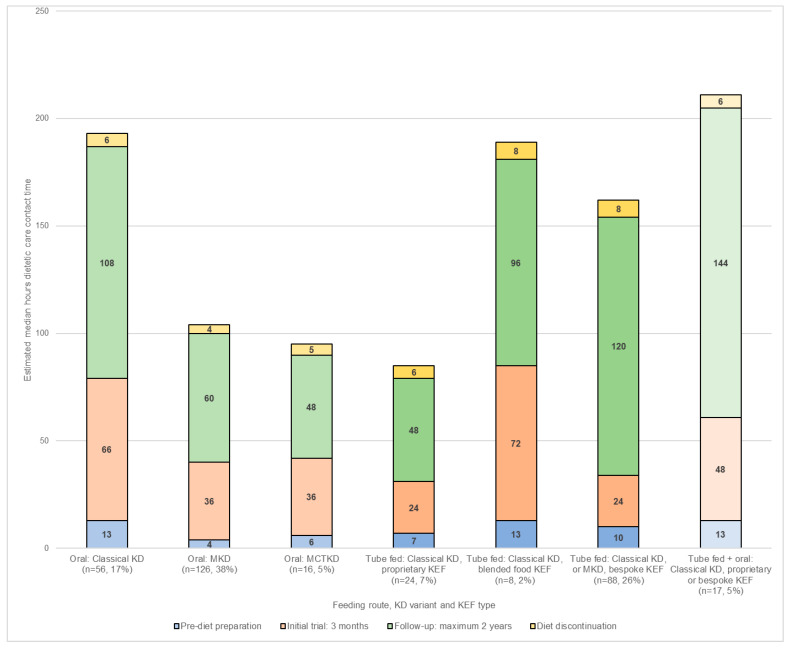
Estimated care contact time (hours) per dietitian, per patient during each of the four stages of the KD patient journey (maximum of 2-years follow-up) by feeding route, KD variant, and ketogenic enteral feed (KEF) type, *n* = 335.

**Figure 2 nutrients-13-02484-f002:**
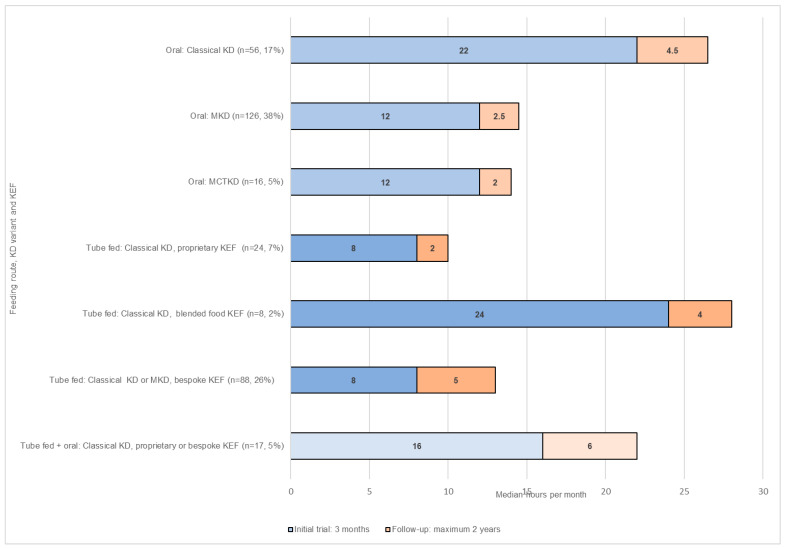
Estimated care contact time (hours) per dietician, per patient, per month during the initial 3-month trial and 2-year follow-up by feeding route, KD variant, and ketogenic enteral feed (KEF) type, *n* = 335.

**Table 1 nutrients-13-02484-t001:** Estimated dietetic contact care time (direct and indirect activity) during the KD patient journey, *n* = 335.

			Stage of Patient KD Journey				
			Pre-Diet Preparation	Initial 3-Month Trial	Follow-up—Maximum of 2 Years (Post 3-Month Initial Trial)	Diet Discontinuation	Total (Including All Four Stages of the KD Patient Journey)
Feeding Route (*n*, %)	KD Variant * (*n*, %)	Type of Ketogenic Enteral Feed (KEF) **	Median Estimated Hours Spent per Patient, per Dietitian (IQR)	Median Estimated Hours Spent per Patient, per Dietitian (IQR)	Median Estimated Hours Spent per Patient, per Dietitian (IQR)	Median Estimated Hours Spent per Patient, per Dietitian (IQR)	Median Estimated Hours Spent per Patient, per Dietitian (IQR)
Oral (*n* = 198, 59%)	Classical KD (*n* = 56, 17%)	n/a	13 (8)	66 (103)	108 (96)	6 (6)	193 (213)
	MKD (*n* = 126, 38%)	n/a	4 (9)	36 (30)	60 (48)	4 (3)	104 (94)
	MCT KD (*n* = 16, 5%)	n/a	6 (5)	36 (36)	48 (36)	5 (5)	95 (82)
Tube (*n* = 120, 36%)	Classical KD (*n* = 24, 7%)	Proprietary	7 (4)	24 (30)	48 (36)	6 (3)	85 (61)
	Classical KD (*n* = 8, 2%)	Blended food	13 (4)	72 (20)	96 (120)	8 (4)	189 (148)
	Classical KD (*n* = 84, 25%)	Bespoke	10 (6)	24 (39)	120 (78)	8 (5)	162 (128)
	and MKD (*n* = 4, 1%)						
Tube + Oral (*n* = 17, 5%)	Classical KD (*n* = 17, 5%)	Proprietary and bespoke	13 (10)	48 (36)	144 (120)	6 (6)	211 (172)
**Median estimated total hours (IQR)**			10 (4)	36 (24)	96 (48)	6 (2)	162 (54)

* KD variant: Classical KD: calculated as a ratio of fat to protein and carbohydrate combined. MCT KD (medium chain triglyceride ketogenic diet): 30–60% of dietary energy provided from MCT. MKD (modified ketogenic diet): as defined by Martin-McGill et al., 2019 [5]. ** KEF type: Proprietary: liquid or reconstituted powdered formulae based on the classical KD and used as is, i.e., no extra ingredients added. Bespoke: modular-ingredient or proprietary ketogenic formula with additional ingredients. Blended food: feed made from real food ingredients.

**Table 2 nutrients-13-02484-t002:** Percentage of dietetic care contact time spent on direct patient activity during the KD patient journey, *n* = 335.

			Stage of Patient KD Journey				
			Pre-Diet Preparation	Initial 3-Month Trial	Follow-up—Maximum of 2 Years (Post 3-Month Initial Trial)	Diet Discontinuation	Total (Including All Four Stages of the KD Patient Journey)
Feeding Route (*n*, %)	KD Variant * (*n*, %)	Type of Ketogenic Enteral Feed (KEF) **	Median % Estimated Time Spent on Direct Patient Activity (IQR)	Median % Estimated Time Spent on Direct Patient Activity (IQR)	Median % Estimated Time Spent on Direct Patient Activity (IQR)	Median % Estimated Time Spent on Direct Patient Activity (IQR)	Median % Estimated Time Spent on Direct Patient Activity (IQR)
Oral (*n* = 198, 59%)	Classical KD (*n* = 56, 17%)	n/a	50 (25)	43 (33)	50 (25)	50 (25)	50 (25)
	MKD (*n* = 126, 38%)	n/a	63 (23)	50 (25)	50 (25)	50 (25)	50 (25)
	MCT KD (*n* = 16, 5%)	n/a	50 (9)	50 (25)	50 (25)	50 (10)	50 (17)
Tube (*n* = 120, 36%)	Classical KD (*n* = 24, 7%)	Proprietary	32 (20)	50 (38)	50 (38)	20 (15)	41 (26)
	Classical KD (*n* = 8, 2%)	Blended food	40 (18)	50 (25)	60 (38)	35 (13)	45 (21)
	Classical KD (*n* = 84, 25%)	Bespoke	40 (18)	45 (38)	50 (38)	50 (25)	48 (31)
	and MKD (*n* = 4, 1%)						
Tube + Oral (*n* = 17, 5%)	Classical KD (*n* = 17, 5%)	Proprietary and bespoke	40 (18)	35 (25)	35 (25)	40 (25)	38 (25)
**Median % of estimated total hours (IQR)**			40 (16)	50 (8)	50 (15)	50 (15)	48 (6)

* KD variant: Classical KD: calculated as a ratio of fat to protein and carbohydrate combined. MCT KD (medium chain triglyceride ketogenic diet): 30–60% of dietary energy provided from MCT. MKD (modified ketogenic diet): as defined by Martin-McGill et al., 2019 [5]. ** KEF type: Proprietary: liquid or reconstituted powdered formulae based on the classical KD and used as is, i.e., no extra ingredients added. Bespoke: modular-ingredient or proprietary ketogenic formula with additional ingredients. Blended food: feed made from real food ingredients.

## Data Availability

The data presented in this study are available on request from the corresponding author.

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
