# Peer review of "Keto-on-the-Clock: A Survey of Dietetic Care Contact Time Taken to Provide Ketogenic Diets for Drug-Resistant Epilepsy in the UK"

_nutrients, 2021, doi:10.3390/nu13082484_

Round 1
Reviewer 1 Report
Although the present study has many limitations and the survey was not validated I found it very interesting as there are almost no data on the subject raised by the Authors. In general the manuscript was very good written, although below I have pointed some inaccuracies which need to be corrected or explained:
- In the Introduction part please add more details regarding parameters of the ketogenic diet and data on the medium time of treatment with this approach.
- The authors report that the patients were looked after by nutritionists, and what about the role of doctors? Have they participated in therapy using a ketogenic diet? There is nothing about it in the manuscript.
- line 216 - on average 162 hours of care per patient, but per what diet duration (median)? This is very important as it might be significant work or not depending on the duration of the ketogenic diet. In Figures 1 and 2 it was assumed that max. duration was 2 years, but what was the medium duration calculated for all patients and in particular sub-groups? It must be clearly stated.
Author Response
- In the Introduction part please add more details regarding parameters of the ketogenic diet and data on the medium time of treatment with this approach.
We are not sure what parameters for the ketogenic diet you would like to see defined here in the introduction, for example, could this be ketosis?
- We have added in ‘designed to induce ketosis’ to line 37
Median time on a ketogenic diet – we have had a look in the literature and been unable to find any specific data on this. So, we would like to propose that we include wording that acknowledges that that the attrition rate for the KD is an issue (from Martin-McGill et al 2020) even though the Kossoff et al 2018 recommendations are for KDs for DRE to be used for a minimum of 3 months and for up to two years.
We have added further information on the recommendations for duration of the KD for DRE before it is discontinued from the consensus report by Kossoff et al 2018 (reference 6):
- Lines 50 – 56: ‘Although attrition from KDs occurs (mainly due to poor diet tolerance or lack of efficacy) [7] and this will impact diet duration, recommendations are that KDs for DRE are discontinued if unsuccessful after a minimum of 3 months, and after two years in patients where they have been efficacious [6]. However, there is no maximum duration for the use of KDs in DRE and 30% of cases continue for longer or return to diet if seizures reoccur after weaning off [6, 9].
- The authors report that the patients were looked after by nutritionists, and what about the role of doctors? Have they participated in therapy using a ketogenic diet? There is nothing about it in the manuscript.
In lines 43 and 44 we refer to the multidisciplinary team (MDT) approach to application of KDs.
Lines 43 and 44: ‘KDs are a resource-intensive treatment, with a dietitian as an essential member of the multidisciplinary team (MDT) [6, 8].
The two references we provide (6 and 8) state that other Health care professionals, for example physicians, provide care for patients on the KD.
We believe we have made it clear throughout the document that our respondents were dietitians and no other health professionals provided information.
Lines 67 – 68 – we state our aim as being an investigation of dietitian’s time only: ‘The aim of this study was to investigate the time taken by dietitians in the UK and Ireland to implement and manage KDs in patients with DRE. The results may be relevant for application in the delivery of dietetic services for DRE’.
However, to clarify we did not consider time taken by other healthcare professionals in our results, we have added this sentence into the Discussion section using this wording:
- Lines 345 - 346: ‘Time required for other healthcare professionals involved in the KD service, such as physicians and clinical nurse specialists, was also not evaluated’.
- line 216 - on average 162 hours of care per patient, but per what diet duration (median)? This is very important as it might be significant work or not depending on the duration of the ketogenic diet. In Figures 1 and 2 it was assumed that max. duration was 2 years, but what was the medium duration calculated for all patients and in particular sub-groups? It must be clearly stated.
Please note that we did not set out to do the survey in this way, i.e. to collect quantitative data on time taken for individual patients but to investigate estimated time taken. If we had asked this question in the survey and collected this data, it would have been a completely different study, and one which we considered was too onerous for potential respondents. This is acknowledged in line 335.
Our aim was to collect data on estimated time taken to manage patients on the KD. In effect, we asked the dietitians to estimate the amount of time they would spend managing a patient for whom the KD was beneficial during a 3-month trial, and then if they were followed up for 2 years. As we did not collect data on the actual time that dietitians spent managing their individual patients, we are unable to provide a median value for diet duration, either for all patients nor the subgroups.
We obtained the number of patients managed on KDs from the dietitians for KD version and feeding route but not data for how long each of them may have been on the diet for. Hence, we are unable to include any data on this in the calculation of the medians for the estimated time. The number of patients represented in the survey on each version and route of the KD is provided on the tables and graphs is for information purposes only.
Therefore, 162 hours is the estimated total median time spent by a dietitian managing a patient with DRE on a KD regardless of KD version or route of administration. Our results are therefore for the time in hours estimated by dietitians to spend managing a patient using the Kossoff et al 2018 recommendations for a minimum of 3 months and for 2 years, presuming the KD is efficacious for management of DRE, and in each stage of the patient journey i.e. introduction, 3 month trial, follow-up and discontinuation.
Reviewer 2 Report
The paper reports on results from a survey among 15 dieticians administering ketogenic diets to mainly pediatric patients with drug resistant epilepsy. The authors evaluate time spent on direct patient care as well as patient treatment related activities, and find differences according to type of diet, type of administration, and patient position in treatment schedule. Furthermore, estimated time needed for patient care and related activities exceeds officially allotted working time. The authors conclude that their data provide a basis to identify potential savings in time and resources.
The paper is well written and generally interesting. Categorization of feeding routine and diet variant is a major strength of the paper. However, I have concerns about the data collection, the data quality and reliability and data analysis. Although the authors readily acknowledge some limitations of their study, fundamental weaknesses affect the conclusions of the paper and cannot simply be argued away.
- As the survey questionnaire is not validated, there is no means to tell if the retrospective estimation of time spent on certain activities is reliable. As the authors state, results from their survey are vastly different from other reports in the literature, which might be due to multiple different factors – but fact is, the authors cannot tell if their online survey even reliably measured what it was supposed to measure. Individual qualitative assessments, such as workload being too high, or patient characteristics leading to increased time commitments might be useful without official validation of the questionnaire. However, the time estimates imply a quantitative accuracy that is not there. I suggest validating the questionnaire in a subgroup of dieticians.
- Who was the population of interest for the online survey? I.e how many dieticians in the KDRN were contacted? How are they distributed across the KD centers? Are the 15 dieticians who took part in the survey representative of all contacted dieticians, e.g. regarding years of experience, number of patients treated, etc?
- Ethics approval: KD centers were asked to seek approval- meaning the authors did not verify that approval was indeed obtained? How exactly did approval look like?
- How was patient privacy ensured, when using free-text fields in the online survey? Especially dealing with pediatric patients?
- Is there information on characteristic of the survey participants? I.e. age, years of experience, number of patients treated?
- The presentation of results (time estimates) need to take into account the cluster structure induced by the treating dietician. I.e. if all n = 17 patients with tube and oral feeding route and classical KD are treated by the same dietician, then you cannot disentangle if time estimates for this patient group are due to patient or dietician characteristics. This needs to be discussed.
Author Response
1. As the survey questionnaire is not validated, there is no means to tell if the retrospective estimation of time spent on certain activities is reliable. As the authors state, results from their survey are vastly different from other reports in the literature, which might be due to multiple different factors – but fact is, the authors cannot tell if their online survey even reliably measured what it was supposed to measure. Individual qualitative assessments, such as workload being too high, or patient characteristics leading to increased time commitments might be useful without official validation of the questionnaire. However, the time estimates imply a quantitative accuracy that is not there. I suggest validating the questionnaire in a subgroup of dieticians.
Response to reviewer: We have acknowledged that our survey was not validated (line 337). We have added to this sentence to include a comment about accuracy, as we are aware this is the case.
Throughout the document we state that the data is estimated to clarify this fact.:
- Line 337: The survey was not validated, and therefore the estimates of time we obtained, although they add to the information available on this subject, cannot be relied upon for accuracy.
We would like to add for your consideration that our survey questionnaire underwent some internal validation amongst the consensus group of dietitians (all members of the KDRN) before it was sent out (Lines 71 and 72, Line 340).
We did not set out to do a full quantitative survey, as this was considered too onerous for potential respondents. This is acknowledged in line 335:
Line 329: ‘Time was estimated retrospectively rather than by a formal prospective workload time analysis. This was a pragmatic approach chosen to place least demand on respondents.
In addition, we would like to add that one reason for doing the survey, albeit using a non-validated tool and obtaining estimates of time taken rather than actual measures, and then to write it up and seek publication was to start to add to the information available in this area, as the perception of dietitians is that some KDs and some patients on KDs take longer to manage than others. So, we see this as a beginning!
2. Who was the population of interest for the online survey?
Line 74: only from KDRN members who were dietitians
3. ie how many dieticians in the KDRN were contacted?
- The survey was sent to all the members of the KDRN. The membership includes dietitians and other health care professionals, e.g., nurses, physicians. The membership email list contains their names only, not their profession, so we do not know how many of the group are dietitians.
4. How are they distributed across the KD centers?
- We do not have the answer to this question. It can be assumed that all 39 KD centres have at least one dietitian in.
- 5. Are the 15 dieticians who took part in the survey representative of all contacted dieticians, e.g. regarding years of experience, number of patients treated, etc?
- We do not have any of this information. Membership of KDRN is open to any dietitian (or other healthcare professional) working with patients on ketogenic diets and to those who work in ketogenic diet related research or industry and is not based on their experience of the KD, or the numbers of patients seen. Therefore, we can only presume the 15 that responded are a heterogenous group. We acknowledge this in the discussion (lines 304 - 306).
6. Ethics approval: KD centers were asked to seek approval- meaning the authors did not verify that approval was indeed obtained? How exactly did approval look like?
Response to reviewer: As professionals, the dietitians were given responsibility for getting approval to participate in the survey for themselves, via an email to their relevant hospital department.
Lines 91 – 92: Dietitians wishing to participate in the survey were advised to seek approval from the local research and audit team at their KD centre via email and retain for their records.
7. How was patient privacy ensured, when using free-text fields in the online survey? Especially dealing with pediatric patients?
We did not ask for any patient data in any of these questions and none was provided in the responses. No patient information was requested (inserted on line 90).
8. Is there information on characteristic of the survey participants? I.e. age, years of experience, number of patients treated?
No, as we have not collected this information. Please see previous response in relation to this question (question 5).
9. The presentation of results (time estimates) need to take into account the cluster structure induced by the treating dietician. I.e. if all n = 17 patients with tube and oral feeding route and classical KD are treated by the same dietician, then you cannot disentangle if time estimates for this patient group are due to patient or dietician characteristics. This needs to be discussed
Differences in estimated dietetic time taken per patient in our study do not take into account the level of expertise of the respondent therefore you cannot disentangle if time estimates are due to the characteristics of the patient or expertise of the dietitian.
We have changed the sentences in lines 303 – 307 to try and reflect this better:
Estimated dietetic time taken per patient will be impacted by the individual patient circumstances or underlying health or social needs, and the experience level or individual practice style of the dietitian. It would be assumed that more complex patients or less experienced dietitians would have increased dietetic time reported, but as we did not collect this data, we cannot confirm this.
Round 2
Reviewer 2 Report
I thank the authors for providing a revised version of the manuscript, and for the time and effort put into the revision. The problems with the underlying data (i.e. no validation of the instrument, no information about the survey participants, no possibility to estimate generalizability/bias) are fundamental. Given the weak underlying data, the authors did the best they could. I’m not sure if there are straightforward ways to improve the paper without tackling the underlying data. Maybe add a paragraph to the limitations about self-selection bias of the participants (i.e. dietitians with higher workload/more time intensive work were more likely to participate)?
Author Response
Thank you for your review of our manuscript and the helpful comments, suggestions, and considerations. We readily accept and acknowledge that our results are flawed due to the non-validation of the instrument we used and the gaps in data that have been exposed in analysing the results and writing up the study. But we hope, as we state in our conclusion, that our study report gives some insight into this subject, and that it may be of interest to dietitians working in this field.
As stated in the aims (lines 67 – 68), our intention was an investigation into estimated time taken, so we originally set out to do just a small project, really for interest. The ketogenic literature contains little information about length of time involved in caring for patients on a KD by dietitians and yet, we believe, it is of significance to them, due to the varied ways in which the KD is implemented and the fact that patients participate in a defined patient journey whilst on the KD (as we described), plus dietitians often say they don’t have enough time in their working day! Our impression was that the perception amongst ketogenic dietitians is that some KDs and certain patient groups on KDs take longer to manage than others, but our question was, is this something that is true or not? The data we have collected would seem to indicate that indeed this is the case. It was felt that for this project, asking for estimates of time taken would be more appropriate rather than conducting the study as a full quantitative survey, which, as stated, was considered too onerous for potential respondents to undertake, although absolutely, it would have been the best way to collect this data, in order for us to have produced more meaningful results.
We have made the following revisions to the manuscript:
Line 330: We fully acknowledge that this study has many limitations.
Lines 335 -= 341: Self-selection bias may have influenced the decision to participate in the survey. We have no knowledge about certain characteristics of the fifteen dietitians that took part as we did not ask for information concerning, for example, level of KD experience, workload, or how many of their patients or associated caregivers required a higher intensity of dietetic input and support. These factors may have also affected reported estimates of time taken for dietetic activities, but we are unable to account for this.
Line 345: cannot be relied upon for accuracy nor can they be generalised.
Lines 355 – 356: Despite its many acknowledged limitations, our survey offers novel insight into the care contact time estimated to be required….
Line 358 – 359: Subject to careful interpretation and application due to the non-validity of the instrument used for data collection, potential bias from respondents, and data inaccuracy, our results may have application in the identification of inefficiencies………
We hope this addresses your concerns with our manuscript and the conduct of the study. Many thanks again.